# Content comparison of unmet needs self-report measures for lymphoma cancer survivors: A systematic review

**Vanessa Boland** [1]*, **Amanda Drury**[2], **Anne-Marie Brady**[1]

**1** School of Nursing & Midwifery, Faculty of Health Sciences, Trinity College Dublin, Dublin 2, Ireland,
**2** School of Nursing, Psychotherapy and Community Health, Faculty of Science & Health, Dublin City University, Dublin 9, Ireland

* vboland@tcd.ie

## Abstract

### Purpose

The increasing recognition of the complex impacts of a cancer diagnosis and its treatment has led to efforts to develop instruments to reflect survivors' needs accurately. However, evidence regarding the content and quality of instruments used to evaluate the unmet needs of lymphoma survivors is lacking. This review aimed to evaluate the psychometric properties and comprehensiveness of available self-report instruments to assess unmet needs and quality of life with adult lymphoma survivors.

### Methods

A systematic search of five databases (CINAHL, EMBASE, Medline, PsycInfo and Scopus) was conducted to identify instruments measuring unmet needs or quality of life outcomes. Original articles reporting the instrument's validation or development via citation screening were retrieved and screened against eligibility criteria. An appraisal of the instrument's measurement properties was conducted, guided by the COSMIN methodology and reported in accordance with PRISMA guidelines. A content comparison using the Supportive Care in Cancer Framework was performed.

### Results

Twelve instruments met the inclusion criteria; only one was explicitly developed for lymphoma (Functional Assessment of Cancer Therapy–Lymphoma). Four instruments focused on the construct of need, and eight focused on quality of life. The psychometric data in the published literature is not comprehensive; there is heterogeneity in their development, content and quality. No included instrument was examined for all COSMIN measurement properties, and methodological quality was variable; all instruments measured at least four domains of need. The emotional domain was reviewed by all instruments (n = 12), and the spiritual and informational domains received the least focus (n = 4 each).

**Data Availability Statement:** All relevant data are within the manuscript and its Supporting Information files.

**Funding:** Vanessa Boland is supported by the School of Nursing and Midwifery, Trinity College

Dublin, PhD Scholarship. The funders had no role in study design, data collection and analysis, decision to publish, or preparation of the manuscript.

**Competing interests:** The authors have declared that no competing interests exist.

## Conclusion

This review provides a platform for instrument comparison, with suggestions for important factors to consider in systematically selecting unmet needs and quality of life self-report measures for adult lymphoma survivors. Considering the various discrepancies and limitations of the available instruments, using more than one instrument is recommended. In selecting measurement instruments, researchers should consider research objectives, study design, psychometric properties and the pros and cons of using more than one measure. Evaluating the participant burden and feasibility of completing the selected instrument is important for lymphoma survivors, a group burdened by cancer-related fatigue and cognitive impairment.

## Introduction

Measurement is central to health research and clinical practice [1]. The centrality of the patient perspective in cancer care is critical to achieving meaningful research outcomes. Patient-reported outcome measures assist in identifying the most prevalent concerns for a target population [2]. Optimal cancer care requires an adequate understanding of the needs of survivors and the factors that influence them [2, 3]. Efforts to improve the availability of care and resources for cancer survivors have been advanced by assessing survivors' needs [4]. As the number of cancer survivors continues to rise, healthcare services and policymakers must understand this population's needs and quality of life outcomes [5]. This information can focus on the most problematic or prevalent concerns for cancer survivors or facilitate the detection of problems that might otherwise be overlooked [6].

Instruments that accurately measure the influence of cancer on quality of life and the level of unmet needs in the expanding population of cancer survivors are needed. Various instruments are available to measure cancer survivors' diverse outcomes, yet guidance on optimising their selection and recommendations for use is scarce. Selecting appropriate instruments for measuring outcomes of interest is challenging given the vast proliferation of such questionnaires over the past decades [7], which have been associated with a general lack of clarity in their intended applications [6, 8].

A cancer diagnosis poses a potential threat to life, disrupts daily functioning and is a barrier to psychological health and well-being [9]. Tumour site-specific research is required to advance future research and healthcare delivery that is responsive to survivors' needs [10, 11]. Lymphoma, the most prevalent haematological cancer, affects an estimated 628,000 new annual cases worldwide across all age groups [12]. Originating in the lymphatic system, the widespread nature of lymphoma can affect any organ in the body, creating a range of symptoms depending on the location [13, 14]. It is categorised into two types, Hodgkin lymphoma (HL) and non-Hodgkin lymphoma (NHL), with indolent and aggressive forms [13].

The varied clinical features and histological appearances of lymphoma present challenges such as difficult diagnosis, complex management strategies and assorted prognoses [15, 16]. Moreover, long-term morbidity and mortality impact quality of life and can pose challenges associated with lymphoma survivorship [16]. Lymphoma survivors are substantially increasing due to success in treatment modalities, producing a growing dependency on health care services from diagnosis and treatment to follow-up and survivorship care [17]. Despite improvements in survival rates, treatment toxicities are endured by patients [18]. Therefore, assessing

needs and quality of life outcomes among people living with and after lymphoma is important to understand better lymphoma survivorship and its influencing factors [19].

Scant guidance is currently available on measuring lymphoma survivors' needs and quality of life outcomes. Numerous quality-of-life instruments are used in the current literature on lymphoma survivors, demonstrating a largely non-standardised approach to measuring different quality-of-life outcomes. This has contributed to the impairment of meta-analytical analysis to form conclusions and recommendations for this population [20]. Moreover, Goswami and colleagues [21] reviewed the quality of life instruments available for haematological malignancies; of the thirty instruments identified, only one was a lymphoma-specific tool (Functional Assessment of Cancer Therapy—Lymphoma).

Conceptual uncertainties exist as to what constitutes a healthcare need. 'Unmet needs' discriminate between the needs experienced by survivors and those they wish for help in managing. 'Quality of life' involves an individual's perception of their position in life, which is context-bound to their culture, values and way of life [22]. The two constructs of interest share an interrelation and are commonly used in cancer survivorship research, a period that begins at diagnosis and continues until the end of life [23, 24]. However, often the factors that have the greatest impact on unmet needs and overall quality of life outcomes are not routinely captured. Thus the most prevalent survivorship issues may not always be the most impactful [25].

An issue for healthcare research is the range of available outcome measurement instruments measuring the same constructs with further patient-reported outcome measures in development. The diversity of available instruments leads to heterogeneity in the evidence generated, which is problematic for evidence-based recommendations. Choosing the most suitable and applicable instrument is vital as selecting inappropriate or poor-quality instruments may present bias in the conclusions of studies [26]. As the availability of generic, cancer-related, and tumour-site-specific outcome measures increases, a review is required to assimilate evidence on available instruments.

This research aims to evaluate the content and quality of instruments used with lymphoma survivors for assessing the constructs of interest (unmet need and quality of life). The Consensus-based Standards for selecting health Measurement Instruments (COSMIN) methodology was developed explicitly for studies on health-related patient-reported outcomes [1] and will support evidence for lymphoma-specific research in this current review. The objectives are:

i. To identify available instruments measuring the unmet needs and quality of life outcomes of lymphoma survivors in current literature.

ii. To conduct a content comparison of the included instruments and appraise the pertinent psychometric measurement properties.

iii. To provide evidence-based recommendations for applying patient-reported lymphoma-specific outcome measurements in future research.

## Methods

### Study design

This systematic review was conducted and reported following the Preferred Reporting Items for Systematic Reviews and Meta-Analysis (PRISMA) Guidelines [27, 28]. The ten-step procedure developed by COSMIN for conducting a systematic review of outcome measures further guides this study [29]. The COSMIN-validated Risk of Bias checklist supports evaluating patient-reported outcome measures, and a user manual provides a step-by-step guide to

instrument appraisal [29–31]. Measurement properties are only assessed if presented in the selected article.

## Search strategy

The authors conducted a recent review using a rapid review methodology and reflexive thematic analysis to gather current evidence on lymphoma survivors' unmet needs and quality of life outcomes [20]. The study focused on adult lymphoma survivors of any subtype or stage. Five databases: CINAHL, EMBASE, Medline, PsycInfo and Scopus, were systematically searched from 2006 –February 2022, limiting the search to English-language articles. A researcher and an information retrieval specialist performed the search strategy and database searches. Two independent reviewers screened and assessed the methodological quality of all included studies. The full review methodology and methods are discussed elsewhere [20].

Articles included in this rapid review and their extracted data (i.e., details on the instruments used) provided a list of instruments relevant to the construct and population of interest. Thirty-six studies included in this review used thirty-one different outcome measurement instruments in a population with at least fifty per cent lymphoma survivors. Next, the published articles reporting the original development or validation of these instruments were retrieved based on the citation(s) provided for each paper. In addition, the COSMIN Database of Systematic Reviews of Outcome Measurement Instruments was searched to cross-check the completeness of identified instruments for this population. The search terms for this database included: "lymphoma", "non-Hodgkin lymphoma", "Hodgkin lymphoma", "haematological malignancy", and "blood cancer". All papers identified via this process were then imported into Covidence for screening according to the inclusion and exclusion criteria of the review.

## Eligibility criteria

The eligibility criteria are outlined in Population, Issue, Comparison, Outcome, Study (PICOS) format in Table 1. Peer-reviewed articles reporting the evaluation of the psychometric properties of relevant self-reported measures for unmet need or quality of life was restricted to studies concentrating on the development or validation of a single instrument. One researcher (V.B.) initially screened instruments, with verification by two other researchers (A.D. and A-M.B.); discrepancies were resolved through discussion and required consensus to minimise selection bias. Studies evaluating an instrument of interest but using a validation process with another instrument were excluded from the review for two reasons: difficulty identifying such articles systematically and interpreting their evidence [1].

## Conceptual framework and content analysis

The Supportive Care Framework for Cancer Care was designed to conceptualise what type of support cancer patients might require and how to approach planning for service delivery [32, 33]. The framework appropriately focuses on distinguishing between concerns experienced by individuals with cancer. Identifying unmet needs offers informed evidence to guide where help is required. The framework's domains encompass the physical, psychological, emotional, social, practical, informational, and spiritual needs of an individual with cancer [32]. Individual mapping of the content of items of the included instruments against the framework's domains was conducted to illustrate the fundamental domains of unmet needs addressed by each instrument.

**Table 1. Eligibility criteria in PICOS format.**

| PEOS | Inclusion Criteria | Exclusion Criteria |
|---|---|---|
| Population | Lymphoma survivors of any stage and subtype. Studies were included if they included lymphoma survivors (>50%) in homogenous or heterogeneous groups. | Populations that are not lymphoma cancer-specific. |
| Issue | Instruments that collect data directly from participants. Content for patient-reported outcomes questionnaires. | Experimental measurements or interventions. Not patient-reported outcomes-based. |
| Comparison | NA | NA |
| Outcome | The instrument should address one of two multidimensional constructs of interest i) unmet need or ii) quality of life. | Constructs that do not meet the definition of needs and quality of life or measure only a single aspect (i.e., anxiety or depression). |
| Study | Development of validation studies on a single instrument. English language only. Full-text reports. | Validation or comparison studies on more than one instrument. Abstracts that need more information to enable adequate data extraction. |

## Assessment of measurement properties

The COnsensus-based Standards for selecting health Measurement Instruments (COSMIN) methodology guided this evaluation [29–31]. If a study meets the standards for good methodological quality, the risk of bias is minimal [1]. The COSMIN taxonomy of measurement properties distinguishes consensus-based definitions of measurement properties and their order of importance. Each present measurement property (structural validity, internal consistency, reliability, measurement error, construct validity, and responsiveness) was evaluated for risk of bias, so the prominent measurement properties identified received appraisal by one reviewer (VB) with verifications by two reviewers (AD and AMB).

# Results

Thirty-one instruments were identified [20]. Several were excluded (n = 19); some were too specific (i.e., Functional Assessment of Chronic Illness Therapy–Fatigue, or Hospital Anxiety and Stress Scale) (n = 9); others had the wrong outcome (i.e., Profile of Mood States) (n = 7) or wrong population (i.e., Support Persons Unmet Needs Survey) (n = 2) (S1 Table). Therefore, twelve instruments were identified across the reviewed studies that met the inclusion criteria: four measured unmet needs and eight measured quality of life. The PRISMA flow diagram (Fig 1) outlines the complete results process.

## Study characteristics

Citation screening identified sixteen original articles reporting on developing and validating the twelve included instruments. Characteristics of the instruments (e.g., purpose, indication) and the study properties (e.g., sample characteristics, country of origin, instrument administration) were extracted to facilitate comparison and analysis. Most instruments (n = 9) had a generic cancer indication; two were generic quality of life measures (SF-36 and EQ-5D-5L) commonly used with cancer survivors [34, 35], and one was lymphoma-specific (FACT-Lym) [18]. Four different instruments focused on the construct of need: Cancer Survivor Unmet Needs (CaSUN), 34-item Short-form Supportive Care Needs Survey (SCNS-SF34), Survivor Unmet Needs Survey (SUNS), and Short-form Survivors Unmet Needs Survey (SF-SUNS). Eight instruments focused on quality of life: EuroQol 5-Dimension, 5-Level instrument (EQ-5D-5L), European Organisation for the Research and Treatment of Cancer Quality of Life Questionnaire (EORTC QLQ-C30), The Functional Assessment of Cancer Therapy–General (FACT-G), The Functional Assessment of Cancer Therapy–Lymphoma (FACT-Lym), Impact

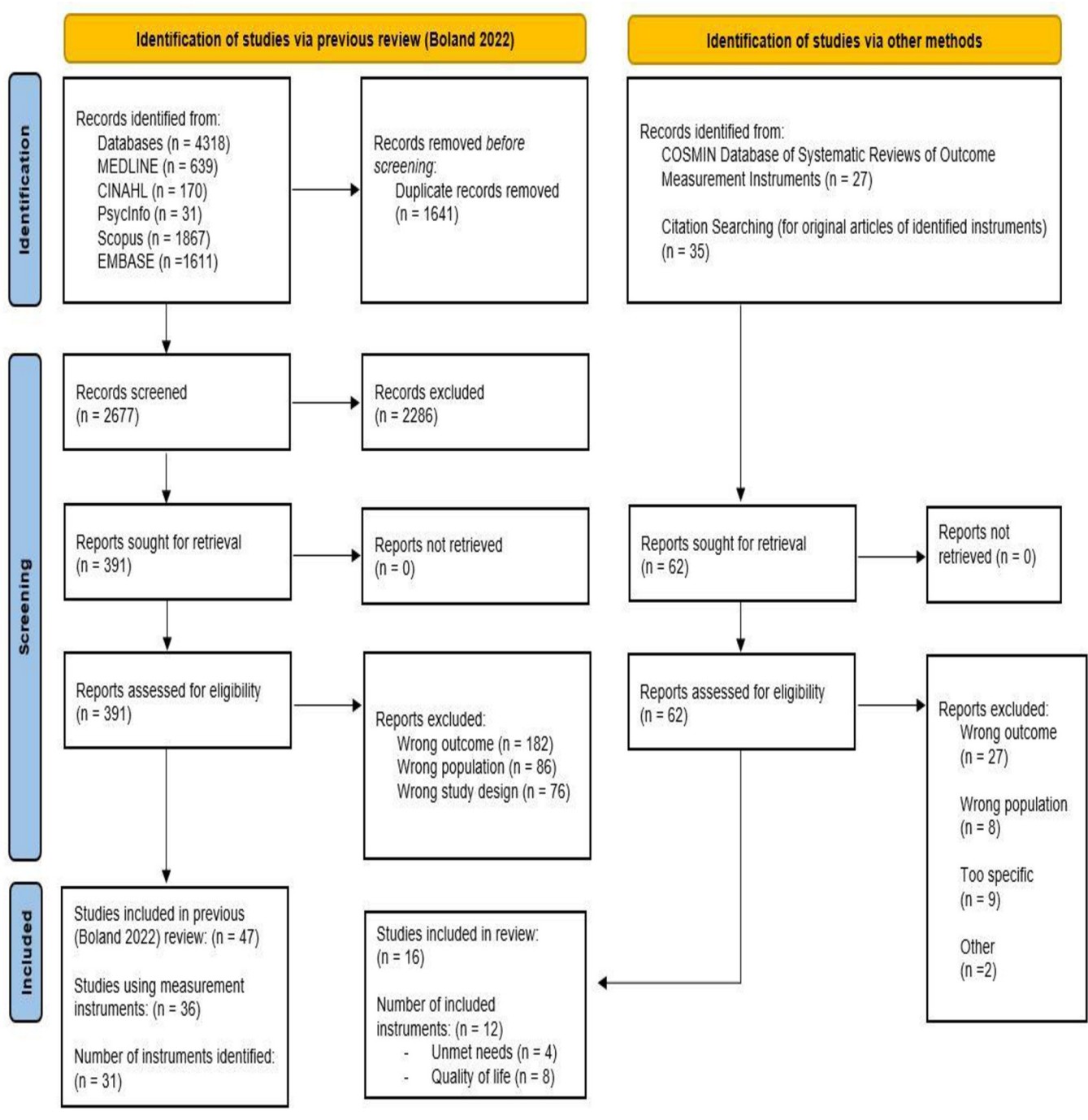

**Fig 1. PRISMA flow chart.**

of Cancer (IOC), Impact of Cancer Version 2 (IOCv2), and Quality of Life–Cancer Survivor (QoL-CS).

The total number of items for instruments varied considerably from five items (EQ-5D-5L and a visual analogue scale) to 89 items (SUNS). The recall period for instruments included the last month (n = 3), the past week (n = 2), and today (n = 1), but some did not clearly define this. Likert scales with five points were mainly used, with a higher score indicating more

problems or a greater need for most instruments (Table 2). The completion time for instrument questionnaires ranged from 5 to 24 minutes (mean = 12 minutes); a completion time was not retrieved for four instruments (IOC, IOCv2, QoL-CS SF-SUNS). Quality-of-life instruments were primarily developed in the U.S.A., with the EORTC QLQ-C30 and EQ-5D-5L developed across multiple countries [35, 36]. Needs instruments originated from Australia (55%) and Canada (45%) [5, 37–39], with one study for the SUNS developed in both countries [40]. Sample sizes of the included studies ranged from 40 to 3,919 participants (Table 3). More than half of the articles (56%) had lymphoma participants in the validation and development studies, with almost 20% using a homogenous lymphoma sample [18, 41, 42]. Two articles recruited participants with heterogeneous haematological cancers [40, 43].

Instruments varied in their targeted survivorship trajectory, including longer-term survivors (CaSUN, IOC, IOCv2 and QoL-CS), more acute (EORTC QLQ-C30, FACT-G and FACT-Lym), and a specified period of one to five years post-diagnosis (SUNS and SF-SUNS). Statistical pooling of results for meta-analysis was not performed as there were not enough studies or sufficiently similar studies to combine results [1]. Therefore, this study's approach to synthesis is the best evidence synthesis. This qualitative analysis considers the methodological quality of studies, the homogeneity of studies, and a content comparison of the instruments [1].

## Content comparison

Comparison of instrument content helps appraise a measure's relevance for a specific purpose. Each domain of the Supportive Care Framework (SCF) by Fitch [33] was addressed by at least four instruments (Table 4).

All of the SCF frameworks domains are covered by the CaSUN. Six instruments (SCNS-SF34, SUNS, SF-SUNS, IOC, IOCv2 and QoL-CS) had greater than 70% coverage with the domains. Four instruments addressed more than half of the domains (EORTC QLQ-C30, FACT-G, FACT-Lym and SF-36), with a generic instrument, the EQ-5D-5L, only covering three domains, physical, emotional and practical. All included instruments addressed the emotional domain, and a majority (n = 11, 92%) covered the practical domain. The physical and social domains were represented by ten instruments each. The item content of seven instruments related to the psychological domain but a minority of instruments (n = 4; 33%) specifically addressed either informational (CaSUN, SCNS-SF34, SUNS and SF-SUNS) or spiritual (CaSUN, IOC, IOCv2, QoL-CS) needs within their items.

## Measurement properties

**Validity.** The validation of psychometric tools ensures that measurements are accurate and meaningful for their target population [44]. Content validity refers to the extent to which an instrument measures what it is intended to measure [45–47]. Content validity was adequately reported by 30% of included studies [5, 35, 37, 48, 49]. For instance, the IOC appropriately reported its concept elicitation in a large qualitative sample of long-term cancer survivors (n = 47) [48]. However, some studies provided limited reporting of qualitative methods undertaken for item generation, while others only utilised quantitative survey methods (S1 Appendix). Short-form instruments usually derive from an original longer-form instrument; the reporting of the newer adapted instrument must reorientate the reader to what the construct is; this is a limitation of the initial reporting for the SCNS-SF34 [38].

A construct is a broad topic for a study; it contains related dimensions (or subscales) treated as a single theoretical concept. Structural validity measures the degree to which the scores of a measurement instrument reflect the construct's dimensionality [50]. The included instruments mainly incorporated a multidimensional and discriminative approach. Confirmatory factor

**Table 2. Summary characteristics of unmet needs instruments (n = 4) and quality of life instruments (n = 8).**

| Instrument | Objective | Indication | Content (total number of items) | Response options | Recall period | Completion time (minutes) |
|---|---|---|---|---|---|---|
| CaSUN | To assess supportive care needs in cancer survivors | Generic cancer | Five subscales: existential survivorship, comprehensive cancer care, information, quality of life, and relationships. Thirty-five unmet need items, six positive change items and an open-ended question **(42 items)** | 5 point (higher score = greater needs) | Last month | 10 |
| SCNS-SF34 | To assess the generic needs of patients with cancer | Generic cancer (all stages) | Five subscales: psychological, health system & information, physical & daily living, patient care & support, and sexuality needs **(34 items)** | 5 point (higher scores = greater need) | Last month | 10 |
| SF-SUNS | A refined version of the SUNS | Generic cancer | Four subscales: information, financial concerns, access & continuity of care, relationships & emotional health **(30 items)** | 5 point (higher scores = greater need) | Last month | Unclear |
| SUNS | To assess the unmet needs of cancer survivors | Generic cancer | Five subscales: informational needs, financial concerns, access & continuity of care, relationships, emotional health **(89 items)** | 5 point (higher scores = greater need) | Last month | 24 |
| EQ-5D-5L | To evaluate the generic quality of life | Generic Quality of life | Descriptive system: 5 dimensions (mobility, self-care, usual activities, pain/discomfort and anxiety/depression) **(5 items** + Visual Analogue Scale) | 1-digit number (1–5) expresses each dimension's severity level (no problems to extreme problems). VAS 0–100 (worst to best health imagined). | Today | < 5 |
| EORTC QLQ-C30 | To assess the quality of life in various cancer patient populations. | Generic cancer | Nine subscales. Five functioning scales: physical, role, emotional, social, and cognitive. Three symptom scales: pain, fatigue, nausea & vomiting. Six items: dyspnoea, appetite, diarrhoea, constipation, financial impact **(30 items)** | 4 points (For physical and role function, higher scores reflect better health. For other items, lower scores reflect better health) | Past Week | 11–12 |
| FACT-G | To measure the quality of life in people with cancer | Generic cancer; chronic disease | Four subscales: physical well-being, social / family well-being, emotional well-being, and functional well-being **(27 items)** | 5 point (higher score = more problems) | Past Week | 5–10 |
| FACT-Lym | To assess the unique issues of a lymphoma diagnosis | Lymphoma | Four subscales: physical wellbeing, social/family wellbeing, emotional wellbeing, functional wellbeing, and additional concerns **(42 items)** | 5 point (higher score = more problems) | Past Week | 10–15 |
| IOCv2 | A refined version of the IOC | Generic cancer | Eight subscales and two summary scores (health awareness, body change, positive and negative self-evaluation, positive and negative life outlook, life interferences, value of relationships, meaning of cancer, and health worry) **(37 items)** | 5 point (higher score = more problems) | Unclear | Unclear |
| IOC | To assess the impact of cancer on the quality of life and other aspects of long-term cancer survivors. | Generic cancer | Ten subscales: health awareness, body changes, health worries, positive and negative self-evaluation, positive and negative life outlook, social life inferences, relationships and meaning of cancer **(81 items)** | 5 point (higher score = more problems) | Some items refer to today | Unclear |
| QoL-CS | To assess the quality of life of long-term cancer survivors | Generic cancer | Four subscales: physical well-being, psychological well-being, social well-being, spiritual/existential well-being **(41 items)** | 10 point (higher scores = better QoL) | Unclear | Unclear |

(*Continued*)

**Table 2.** (Continued)

| Instrument | Objective | Indication | Content (total number of items) | Response options | Recall period | Completion time (minutes) |
|---|---|---|---|---|---|---|
| SF-36 | To measure the generic quality of life | Generic quality of life | Eight subscales: physical functioning, mental health, role physical, role emotional, social functioning, bodily pain, general health perceptions and vitality **(36 items)** | Varies with question | Varies (now, past four weeks) | < 10 |

CaSUN = Cancer Survivor Unmet Needs, SCNS-SF34 = 34-item Short-form Supportive Care Needs Survey, SF-SUNS = Short-form Survivors Unmet Needs Survey, SUNS = Survivor Unmet Needs Survey, EQ-5D-5L = EuroQol 5-Dimension, 5-Level instrument, EORTC QLQ-C30 = European Organisation for the Research and Treatment of Cancer Quality of Life Questionnaire, FACT-G = The Functional Assessment of Cancer Therapy–General, FACT-LYM = The Functional Assessment of Cancer Therapy–Lymphoma, IOC = Impact of Cancer, IOCv2 = Impact of Cancer Version 2 and QoL-CS = Quality of Life–Cancer Survivor.

analysis is preferred over explorative factor analysis when a strong theory about the instrument's structure is known, although both are useful in evaluating structural validity [31]. Several studies (44%) clearly described the use of exploratory factor analysis; its overarching goal is to identify the underlying relationships between measured variables [5, 37, 39, 41, 43, 48, 51].

**Reliability.** Reliability involves the degree to which the measurement is free from error [50, 52]. Test-retest reliability measures the consistency of the same test over time [50] and helps us understand how dependable a measurement instrument is. If the correlation between the results at different time points is high, this is considered evidence for good test-retest reliability. Intraclass correlation coefficients describe how strongly units in the same group resemble each other [53]. Intraclass coefficients for all subscales of the following instruments were calculated and inferred that the reliability for the FACT-G (0.81–0.92) was excellent, FACT-Lym (0.61–0.87) was good to excellent, and SF-SUNS (0.45–0.74) was fair to good [18, 42, 54]. The Kappa (k) statistic's strength of agreement ranges from <0.40 (fair), 0.41–0.60 (moderate), 0.61–0.80 (substantial) and >0.81 (almost perfect) and provides a benchmark for interpretation [53]. The test-retest reliability was low for the CaSUN (mean k = 0.13) and moderate for the SUNS (mean k = 0.58). While statistical outputs, as outlined above, are reported as verifications, the methods used to provide evidence for this reliability are mostly insufficiently explained to validate their use or evidence in the context of a wider population (i.e., no confidence intervals).

Internal consistency relates to how well an instrument measures what it aims to measure, or in other words, the degree of interrelatedness among items [50]. Internal consistency only requires one data set (i.e., calculated without repeating the test) and is assessed by Cronbach's alphas which quantifies the level of agreement on a standardised scale (0 to 1). Higher values show higher agreement between items. Excellent internal consistency statistics ($\alpha = >0.70$) were found in all subscales for half of the included studies; this indicates a high degree of homogeneity among the items with respective instruments [5, 18, 38, 39, 43, 49, 51].

**Other measurement properties.** The prominent measurement properties of these instruments have been outlined above. However, other aspects warrant consideration, such as responsiveness, which aims to detect change over time in the construct to be measured. A barrier to assessing responsiveness is the need for more clarity about this measurement property in the literature [1]. One key aspect of responsiveness is its application; it is relevant for measurement instruments using an evaluative application (i.e., a longitudinal study measuring change over time) [1]. Many of the included studies in this review discriminated between participants at a single time-point, and therefore responsiveness is not an issue.

**Table 3. Summary characteristics of the original development studies for unmet needs instruments (n = 7) and quality of life outcomes instruments (n = 9).**

| Instrument | Original development study (author, year) | Population | Sample characteristics (% lymphoma) | Disease duration | Survivorship stage | Instrument administration | Country |
|---|---|---|---|---|---|---|---|
| CaSUN | Hodgkinson (2007) | n = 353 | Heterogenous cancer survivors | Diagnosed one or more years earlier and disease-free | 1 to 15 years post-diagnosis | Two hospital outpatient clinics (breast cancer, mixed cancers) | Australia |
| SCNS-SF34 | Boyes (2009) | n = 888 | Heterogenous cancer survivors | Unclear | Unclear | Secondary analysis from nine cancer treatment centres in one Australian state | Australia |
| SF-SUNS | Campbell (2014) | n = 1,589 | Heterogenous cancer survivors (**5.3% NHL**) | 12 to 60 months post-cancer diagnosis | 1 to 5 years post-diagnosis | Three population-based cancer registries | Canada |
| | Taylor (2018) | n = 40 | Homogenous lymphoma patients (**72.5% NHL, 27.5% HL**) | Three months post-treatment completion | 1 to 5 years post-diagnosis | One tertiary hospital in Australia | Australia |
| SUNS | Campbell (2011) | n = 550 | Heterogenous cancer survivors | 12–60 months post-cancer diagnosis | 1 to 5 years post-diagnosis | Population-based cancer registry | Canada |
| | Hall (2013) | n = 437 | Heterogenous haematological cancer survivors (>**50% lymphoma**) | 1–60 months post-cancer diagnosis | 1 to 5 years post-diagnosis | Australian and Canadian cancer registries | Australia, Canada |
| | Hall (2014) | n = 715 | Heterogeneous haematological cancer survivors (**59% NHL, 6.2% other lymphoma**) | Median time since diagnosis was 35 months | 1 to 5 years post-diagnosis | Four Australian state population-based cancer registries | Australia |
| EQ-5D-5L | Janssen (2013) [64] | n = 3,919 | Patients with chronic conditions | NA | NA | Various administrations in 5 countries | Denmark, England, Italy, the Netherlands, Poland and Scotland |
| EORTC QLQ-C30 | Aaronson (1993) | n = 305 | Homogenous nonresectable lung cancer | Unclear | Unclear | Health centres in 13 countries | USA |
| FACT-G | Cella (1993) | n = 854 | Heterogenous cancer patients (**8% lymphoma and leukaemia**) | Unclear | Unclear | Medical centre, support centre and from an intervention study | USA |
| FACT-Lym | Hlubocky (2013) | n = 84 | Homogenous NHL patients (**100% NHL**) | At least two months after diagnosis of NHL | Unclear | Three medical centres in one city | USA |
| IOCv2 | Crespi (2008) | n = 1,840 | Breast cancer survivors and non-Hodgkin lymphoma survivors (**35% NHL**) | Five to ten years post-diagnosis, disease-free | Long-term survivors | Cancer registries, members of the previous study | USA |
| | Crespi (2010) | n = 652 | Homogenous NHL survivors (**100% NHL**) | At least two years post-diagnosis | Long-term survivors | Cancer registries | USA |
| IOC | Zebrack (2006) | n = 193 | Heterogenous long-term cancer survivors (**25% lymphoma**) | Five to ten years post-diagnosis, disease-free, off treatment | Long-term survivors | One university cancer registry | USA |
| QoL-CS | Ferrell (1995) | n = 686 | Heterogenous cancer patients (**9% lymphoma, 8% HL**) | 4 to 538 months post-diagnosis | Long-term survivors | Cancer survivorship mailing list | USA |

(*Continued*)

**Table 3.** (Continued)

| Instrument | Original development study (author, year) | Population | Sample characteristics (% lymphoma) | Disease duration | Survivorship stage | Instrument administration | Country |
|---|---|---|---|---|---|---|---|
| SF-36 | McHorney (1994) [65] | n = 3,445 | Patients with chronic medical and psychiatric conditions | NA | NA | Various health centres in three cities | USA |

CaSUN = Cancer Survivor Unmet Needs, SCNS-SF34 = 34-item Short-form Supportive Care Needs Survey, SF-SUNS = Short-form Survivors Unmet Needs Survey, SUNS = Survivor Unmet Needs Survey, EQ-5D-5L = EuroQol 5-Dimension, 5-Level instrument, EORTC QLQ-C30 = European Organisation for the Research and Treatment of Cancer Quality of Life Questionnaire, FACT-G = The Functional Assessment of Cancer Therapy–General, FACT-LYM = The Functional Assessment of Cancer Therapy–Lymphoma, IOC = Impact of Cancer, IOCv2 = Impact of Cancer Version 2 and QoL-CS = Quality of Life–Cancer Survivor.

Cross-cultural validity should be considered when an instrument is used in culturally different populations (i.e., ethnicity, language) and refers to the performance of a culturally adapted instrument [50]. For example, one study found that haematological survivors (greater than half had a lymphoma diagnosis) from Australia and Canada responded similarly to items on the SUNS using logistic regression analysis [40]. However, few of the included tools have been subject to cross-cultural testing.

## Discussion

To provide optimal supportive cancer care, identifying patients' perceived concerns and the level of support needed is required [39]. Standardisation in selecting outcome measurement instruments in specific research areas is warranted as it contributes to improved consistency in reporting, enables comparisons, and synthesises findings [26]. The overarching purpose of this review was to address the gap in current knowledge for adult lymphoma survivors by appraising and comparing the available unmet needs and quality of life self-report measures for use with this population. As the number of lymphoma survivors continues to rise, so does their dependency on health systems and services. However, current literature for this population is disparate, and only one of the twelve included instruments was explicitly designed for use with lymphoma survivors (FACT-Lym).

**Table 4. Supportive care framework's domains addressed by each included instrument (Fitch 2000, 2008).**

| Domains vs Instruments | CaSUN | SCNS-SF34 | SUNS | SF-SUNS | EQ-5D-5L | EORTC QLQ-C30 | FACT-G | FACT-Lym | IOC | IOCv2 | QoL-CS | SF36 | % Selected |
|---|---|---|---|---|---|---|---|---|---|---|---|---|---|
| Physical | X | X | | | X | X | X | X | X | X | X | X | 83 |
| Psychological | X | X | X | X | | | | | X | X | X | | 58 |
| Emotional | X | X | X | X | X | X | X | X | X | X | X | X | 100 |
| Social | X | | X | X | | X | X | X | X | X | X | X | 83 |
| Practical | X | X | X | X | X | X | X | X | X | X | | X | 92 |
| Informational | X | X | X | X | | | | | | | | | 33 |
| Spiritual | X | | | | | | | | X | X | X | | 33 |
| % Covered | 100 | 71 | 71 | 71 | 43 | 57 | 57 | 57 | 87 | 87 | 71 | 57 | |

Several instruments were found to measure unmet needs and quality of life outcomes in this population. This implies a need for more consensus on the most suitable instrument for this group. This heterogeneity of instruments makes conducting a comparison of studies challenging. Furthermore, it poses limitations for international research efforts focused on improving or responding to the needs of lymphoma survivors. The principal psychometric measurement properties reported in the reviewed studies are structural validity, internal consistency, reproducibility, and content validity (S1 Appendix). The psychometric data in the published literature needs to be more comprehensive, and the validity and reliability of evidence for available needs assessment tools are limited [55–57].

Similar to other systematic reviews on instrument selection, the included instruments for this review were not examined for all psychometric properties, and evidence for responsiveness was scarce [58, 59]. The COSMIN methodology aims for more standardisation in using outcome measurement instruments [26]. Improved reporting of the aims and related methods is required to permit a greater level of interpretation regarding the precision and the applicability of results in instrument development. Thus, providing a better indication of the suitability and strength of the results.

It was difficult to discern conclusively which measure was the most valid and reliable given that no study assessed all psychometric properties with variation in the psychometric properties they selected to assess. However, the tools with the most comprehensively reported psychometric properties were the SUNS and SF-SUNS [5, 39]. All included unmet needs instruments were developed in Australia or Canada. The FACT-G and its module for lymphoma, FACT-Lym, have been identified for lymphoma survivors. The original study for its development was conducted with a non-Hodgkin lymphoma sample [18]. All included quality of life instruments were developed in the USA except one (EQ-5D-5L). Therefore, the psychometric testing for included instruments outside the countries they were developed is limited. Cultural relevance is an important factor in instrument selection as popularity may not consider the sustainability of the instrument's relevance as treatments and models of care evolve, and new technology becomes available. The awareness surrounding the cultural relevance of an instrument and the cultural nuances which might affect cancer survivors' capacity to communicate their needs and quality of life via self-reported outcome measures requires attention [60].

The acceptability of the selected instrument to the study's population stage of survivorship should be considered. There are several options for long-term survivorship (IOC, IOCv2, QoL-CS, and CaSUN), while the SUNS and SF-SUNS are specific to one to five years post-diagnosis. Cancer survivors are prone to fatigue and impaired cognitive functioning. Therefore, the feasibility of instruments (i.e., fewer items and shorter time to completion) should receive prominent attention; the EQ-5D-5L, EORTC QLQ-C30 and SF-SUNS are the shortest included instruments with thirty or fewer items. By using shorter instruments, there may be fewer participant burdens, potentially a higher response rate, and fewer missing values [61]. There was significant variation between the number of items and domains across instruments. Half of the included quality-of-life instruments were developed in the 1990s, compared to the development of unmet needs instruments published from 2007 onwards. The reporting of unmet needs instruments was more comprehensive than the quality of life instruments, allowing more clarity in the extraction of the aims and methods of psychometric analysis. The domains of informational needs and spiritual needs received the least attention, with the CaSUN having the most coverage of domains of need [37, 62].

In the reviewed literature, statistical tests are common to evaluate reliability. However, the rationale for selecting specific reliability tests and assumptions and parameters underpinning the selected test often needs to be stated, representing a significant limitation of the literature. Statements like 'reliability were assessed using intraclass correlation coefficient' are made

without noting important distinctions in its assumptions and evidence for its selection to facilitate appropriate interpretation. For instance, vague statements are made on the range of results rather than the specific subscale and overall scale results [29].

The COSMIN methodology provides comprehensive resources to assess instruments' psychometric measurement properties [29, 50]. These resources are significant for researchers developing research projects and evaluating their choice of the measurement instrument. However, barriers to its implementation exist, including the time, skill and knowledge base required to assess the COSMIN checklist [63]. The checklist has 114 items; ninety-six are related to psychometric properties; thus, considerable time is required to become acquainted with the COSMIN taxonomy and tools. Its use is a complex activity with limited or not sufficiently similar studies to enable the effective combination of results (i.e., statistical pooling). The complexity of this tool may limit the application and use of COSMIN to compare instruments across diverse health issues. An exploration of how the tool has been applied to date is recommended to establish whether there are opportunities to develop short-form versions of the tool to support rapid decision-making for tools in clinical environments.

## Limitations

While efforts were made to comprehensively review all relevant instruments for the population and outcomes of interest, this review is not exhaustive. Instruments may have extensive validation within studies not detected in this review. The review was limited to English language publications, and instruments in other languages (i.e., studies examining cross-cultural acceptability) may have been missed. A prospective protocol registration for this review was not conducted. However, a rigorous and systematic approach was conducted to identify instruments and evaluate their content and psychometric properties for lymphoma cancer patients and survivors.

## Conclusion

Standardisation in selecting outcome measurement instruments in specific research areas is warranted as it contributes to improved consistency in reporting and reduces difficulties in comparing and synthesising findings [26]. There is a continued need to work in partnership with lymphoma survivors to ensure that future care is responsive to the concerns of this population. Like other tumour-site-specific research, lymphoma survivors, a substantially increasing cohort, will benefit from focused research. Healthcare professionals endeavour to ensure that physical or mechanical tools (i.e., a sphygmomanometer) provide accurate information with each use (i.e., correct blood pressure reading). This concern should be granted to using measurement instruments about their validity and reliability in research and broader clinical practice. However, selecting outcome measures for specific purposes is complex, involving conceptual considerations (i.e., defining the construct and population), practical aspects (i.e., the burden for patients and costs) and quality aspects assessed by different measurement properties. Further studies are warranted to assess the measurement properties of existing instruments, especially for structural validity, cross-cultural validity and reliability.

This review provides a platform for instrument comparisons for adult lymphoma survivors, with suggestions for important factors to consider in systematically selecting unmet needs and quality of life self-report measures. Primarily focused on approaches for research, this review has provided steps to consider for clinical applications. In selecting measurement instruments, researchers should consider research objectives, study design, psychometric properties, feasibility, and the pros and cons of using more than one measure.

## Supporting information

**S1 Table. Excluded instruments.**
(DOCX)

**S1 Checklist. PRISMA checklist.**
(DOCX)

**S1 Appendix. The reliability and validity of unmet needs and quality of life instruments used with adult lymphoma survivors.**
(DOCX)

## Author Contributions

**Conceptualization:** Vanessa Boland, Amanda Drury, Anne-Marie Brady.

**Data curation:** Vanessa Boland, Amanda Drury, Anne-Marie Brady.

**Formal analysis:** Vanessa Boland, Amanda Drury, Anne-Marie Brady.

**Investigation:** Vanessa Boland.

**Methodology:** Vanessa Boland, Amanda Drury, Anne-Marie Brady.

**Project administration:** Vanessa Boland, Amanda Drury, Anne-Marie Brady.

**Resources:** Vanessa Boland, Amanda Drury, Anne-Marie Brady.

**Software:** Vanessa Boland.

**Supervision:** Amanda Drury, Anne-Marie Brady.

**Validation:** Vanessa Boland, Amanda Drury, Anne-Marie Brady.

**Visualization:** Vanessa Boland.

**Writing – original draft:** Vanessa Boland, Amanda Drury, Anne-Marie Brady.

**Writing – review & editing:** Vanessa Boland, Amanda Drury, Anne-Marie Brady.

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
