## [Decision Letter · Decision Letter 0]

26 Jun 2023

PONE-D-23-02229Content comparison of unmet needs self-report measures for lymphoma cancer survivors: A systematic reviewPLOS ONE

Dear Dr. Boland,

Thank you for submitting your manuscript to PLOS ONE. After careful consideration, we feel that it has merit but does not fully meet PLOS ONE’s publication criteria as it currently stands. Therefore, we invite you to submit a revised version of the manuscript that addresses the points raised during the review process.

The reviewers suggested minor revision.

Please do modifications according to the suggestions.

We look forward to receiving your revised manuscript.

Best regards,

Dr Antonino Maniaci

Academic Editor

PLOS ONE

Journal Requirements:

2. Please include your tables as part of your main manuscript and remove the individual files. Please note that supplementary tables (should remain/ be uploaded) as separate "supporting information" files

Reviewers' comments:

Reviewer's Responses to Questions

**Comments to the Author**

1. Is the manuscript technically sound, and do the data support the conclusions?

Reviewer #1: Yes

Reviewer #2: Yes

2. Has the statistical analysis been performed appropriately and rigorously? 

Reviewer #1: N/A

Reviewer #2: Yes

3. Have the authors made all data underlying the findings in their manuscript fully available?

Reviewer #1: Yes

Reviewer #2: Yes

4. Is the manuscript presented in an intelligible fashion and written in standard English?

Reviewer #1: Yes

Reviewer #2: Yes

5. Review Comments to the Author

Reviewer #1: I read with great interest the manuscript by Boland et al. on self-report measures for lymphoma cancer survivors. The paper is interesting and original. However, I have some minor issues to be addressed:

Introduction

- You should add that several instruments have been studied to increase quality of life of patients with cancer in different settings (doi: 10.3390/nu14142958 - doi: 10.1007/s00520-020-05869-0), expecially during COVID-19 pandemic (https://doi.org/10.1186/s44158-022-00067-2). Please briefly discuss and add these 3 references.

Methods

- All systematic reviews should be registered in PROSPERO before starting the systematic search. Did you perform a registration? Please insert the registration number or add it as a limitation of the study.

- Please modify table 1 into PICOS format (intervention and control must be underlined).

Results

- Please explain in the results section why a meta-analysis was not feasible.

Discussion

- Please add a conclusion section

Reviewer #2: Introduction

Emphasize the importance of measuring cancer survivors' unmet needs and quality of life outcomes.

Clarify the focus on lymphoma cancer survivors and the need for tumor site-specific research.

Explain the challenges in selecting appropriate instruments for measuring outcomes.

State the objectives of the systematic review.

Precision treatment of post-pneumonectomy unilateral laryngeal paralysis is crucial for patients who have undergone cancer or thyroid surgery. This approach involves targeted therapies and interventions, such as laryngeal reinnervation or vocal fold injections, to restore vocal fold function and improve patient outcomes. Tailoring the treatment to each individual's specific needs enhances the effectiveness and ensures optimal recovery of vocal function and quality of life.please discuss and cite doi:10.2217/fon-2019-0053

Methods

Organize the methods section with subheadings for study design, search strategy, eligibility criteria, conceptual framework and content analysis, and assessment of measurement properties.

Simplify descriptions of the search strategy and eligibility criteria.

Describe the Supportive Care Framework for Cancer Care and its application in the content analysis.

Results

Present a summary of the identified instruments and their psychometric properties.

Discuss the content comparison of instruments based on the Supportive Care Framework for Cancer Care.

Provide evidence-based recommendations for selecting appropriate instruments for future research.

Discussion

Vocal outcomes after cervical surgery, particularly thyroid and parathyroid procedures, can vary depending on the surgical techniques employed and the preservation of the recurrent laryngeal nerve. Minimally invasive and nerve-sparing approaches are often utilized to reduce the risk of vocal fold paralysis and voice dysfunction. Meticulous surgical planning, intraoperative neuromonitoring, and surgeon expertise play crucial roles in ensuring optimal vocal outcomes and minimizing complications postoperatively, please discuss and cite DOI:10.23812/19-282-L

Conclusion

Summarize the findings and their implications for lymphoma cancer survivor research.

Emphasize the importance of selecting appropriate instruments for measuring unmet needs and quality of life outcomes.

Suggest areas for future research and improvement in the field.

6. PLOS authors have the option to publish the peer review history of their article (what does this mean?). If published, this will include your full peer review and any attached files.

Reviewer #1: No

Reviewer #2: No

---

## [Author Response · Author response to Decision Letter 0]

25 Jul 2023

Detailed response to reviewers provided in file 'Responses to Reviewers'. Many thanks for your time, effort and expertise in this review process.

---

## [Editor Report · Decision Letter 1]

15 Aug 2023

Content comparison of unmet needs self-report measures for lymphoma cancer survivors: A systematic review

PONE-D-23-02229R1

Dear Dr. Boland,

We’re pleased to inform you that your manuscript has been judged scientifically suitable for publication and will be formally accepted for publication once it meets all outstanding technical requirements.

Kind regards,

Antonino Maniaci

Academic Editor

PLOS ONE

Additional Editor Comments (optional):

The paper improved After the suggestions required and modifications.

Best
---

## [Editor Report · Acceptance letter]

23 Aug 2023

PONE-D-23-02229R1 

Content comparison of unmet needs self-report measures for lymphoma cancer survivors: A systematic review 

Dear Dr. Boland:

I'm pleased to inform you that your manuscript has been deemed suitable for publication in PLOS ONE. Congratulations! Your manuscript is now with our production department. 

Kind regards, 

on behalf of

Dr. Antonino Maniaci 

Academic Editor

PLOS ONE